# Differential responses to folic acid in an established keloid fibroblast cell line are mediated by JAK1/2 and STAT3

**Katelyn J. McCann**[1], **Manoj Yadav**[1,2], **Mohammadali E. Alishahedani**[1,2], **Alexandra F. Freeman**[1], **Ian A. Myles**[1,2]*

1 Laboratory of Clinical Immunology and Microbiology, NIAID, NIH, Bethesda, Maryland, United States of America, 2 Epithelial Therapeutics Unit, Laboratory of Clinical Immunology and Microbiology, National Institute of Allergy and Infectious Diseases, National Institutes of Health, Bethesda, Maryland, United States of America

* mylesi@niaid.nih.gov

**Data Availability Statement:** All relevant data are within the paper and its Supporting information files.

## Abstract

Keloids are a type of disordered scar formation which not only show heterogeneity between individuals and within the scar itself, but also share common features of hyperproliferation, abnormal extra-cellular matrix deposition and degradation, as well as altered expression of the molecular markers of wound healing. Numerous reports have established that cells from keloid scars display Warburg metabolism—a form of JAK2/STAT3-induced metabolic adaptation typical of rapidly dividing cells in which glycolysis becomes the predominant source of ATP over oxidative phosphorylation (OxPhos). Using the JAK1/2 inhibitor ruxolitinib, along with cells from patients with STAT3 loss of function (STA3 LOF; autosomal dominant hyper IgE syndrome) we examined the role of JAK/STAT signaling in the hyperproliferation and metabolic dysregulation seen in keloid fibroblasts. Although ruxolitinib inhibited hyperactivity in the scratch assay in keloid fibroblasts, it paradoxically exacerbated the hyper-glycolytic state, possibly by further limiting OxPhos via alterations in mitochondrial phosphorylated STAT3 (pSTAT3$^{Ser727}$). In healthy volunteer fibroblasts, folic acid exposure recapitulated the exaggerated closure and hyper-glycolytic state of keloid fibroblasts through JAK1/2- and STAT3-dependent pathways. Although additional studies are needed before extrapolating from a representative cell line to keloids writ large, our results provide novel insights into the metabolic consequences of STAT3 dysfunction, suggest a possible role for folate metabolism in the pathogenesis of keloid scars, and offer *in vitro* pre-clinical data supporting considerations of clinical trials for ruxolitinib in keloid disorder.

## Introduction

Keloids are a type of disordered scar formation characterized by hyperproliferation, abnormal extra-cellular matrix deposition and degradation, as well as altered expression of wound healing molecular markers [1–3]. Although keloid scars have common aesthetic and physical consequences, the disorder is highly heterogeneous, demonstrating: population-level

**Funding:** This work was supported by the Intramural Research Program of NIAID and the NIH.

**Competing interests:** The authors have declared that no competing interests exist.

heterogeneity between keloids on different individuals; topographical heterogeneity between body sites on the same individual; as well as cellular and molecular heterogeneity within the scar itself [1]. Numerous reports have established that cells from keloid scars display Warburg metabolism [4–7]. First described in neoplastic cells by Dr. Otto H. Warburg [8], the Nobel winning discovery identified the propensity for rapidly dividing cells to preferentially utilize glycolysis for ATP generation at the expense of oxidative phosphorylation (OxPhos) despite available oxygen [9]. Subsequent research identified over activity of signal transducer and activator of transcription 3 (STAT3; in conjunction with JAK2 [10]) as a central mediator of keloid pathology [5, 11]. Furthermore, recent publications in cancer cells have identified the JAK/STAT signaling pathways as an inducer of Warburg metabolism [12, 13].

We previously demonstrated the importance of STAT3 in the scratch assay, an *in vitro* model of epithelial to mesenchymal transition (EMT; a central process of proliferation and migration of epithelial cells) mediated wound healing in patients with autosomal dominant hyper IgE syndrome due to STAT3 loss of function (STAT3LOF) [14]. We additionally demonstrated a role for STAT3 in the therapeutic response to the commensal flora *Roseomonas mucosa* in atopic dermatitis [15]. The influence of STAT3 on Warburg physiology as well as on cell proliferation and migration require concurrent activation of JAK2 [5, 12, 16]. Given that both STAT3 and JAK2 are functions are abnormal in keloid derived cells [5, 16], we hypothesized that pharmacologic and mutational disruptions of JAK2/STAT3 would influence both the metabolic and proliferation/migration abnormalities seen in keloid cells.

Although the JAK1/2 inhibitor ruxolitinib inhibited the hyperactivity of a representative keloid fibroblast cell line in the scratch assay, it worsened the hyper-glycolytic state. While the keloid fibroblast line had greater total phosphorylated STAT3 (pSTAT3$^{Ser727}$), it had less pSTAT3$^{Ser727}$ co-localized with the mitochondrial marker TOM70, suggesting that STAT3-mediated abnormalities may be influenced by the mitochondrial targeting of pSTAT3. In a representative healthy volunteer fibroblast line, folic acid exposure recapitulated the exaggerated wound closure and hyper-glycolytic state of keloid fibroblasts through JAK1/2-dependent pathways. Although additional studies are needed before extrapolating from a representative cell line to keloids writ large, our results add to the growing literature implicating the JAK/STAT3 pathway in the pathogenesis of keloid scars and suggest a possible role for folate metabolism in the pathogenesis of keloid scars and offer pre-clinical data supporting consideration of clinical trials investigating the use of ruxolitinib in keloid disorder.

## Methods

All work was approved by the Institutional Review Board of the National Institutes of Health.

### Cell cultures and scratch assay

The human primary fibroblast cell line (ATCC PCS-201-012) and the human primary human keloid fibroblast cell line (ATCC CRL-1762) were purchased from American Tissue Culture Collection (ATCC; Manassas). Fibroblasts from patients with STAT3 loss of function were a kind gift from the Boehm lab (NIH, Bethesda, MD) collected under the IRB approved clinical trial NCT00006150 after collecting both written and oral consent from each subject. All cells were cultured and proliferated as previously described [14]. 96 or 24 well plates (Corning; Corning, NY) were coated with 1mg/mL rat tail collagen (Roche; Basel, Switzerland) overnight at 4°C. For 96 well plates, 17,000–25,000 cells and for 24-well plates 100,000–150,000 cells were seeded and allowed to adhere to the culture plate for 1–2 hours. Challenge with drugs or metabolites occurred for 1 hour prior to scratch using the Autoscratch (BioTek; Winooski, VT), or 2 hours prior to Seahorse (discussed below). Cells were placed in the Cytation 5

(BioTek) at 37°C with 5% CO2; images and quantitation were performed by the Scratch App (BioTek). Ruxolitinib was purchased from Caymen Chemicals (Ann Arbor, MI). Folic acid was purchased from Sigma Aldrich (St. Louis, MO). 2-Deoxy-D-glucose (2DG) was purchased from Sigma Aldrich (St. Louis, MO) and cell were treated with a final concentration of 1mM. Rotenone was purchased from Sigma Aldrich (St. Louis, MO) and cell were treated with a final concentration of 1μM. Cells were used between passages 2 and 10, with matching passage numbers used within any given experiment.

## Immunofluorescence staining

Cells were fixed with 4% Paraformaldehyde solution (PFA) (Cat. No. 15710; Electron Microscopy Sciences, Hatfield, PA) for 20 minutes. After fixation cells were processed for the immunostaining protocol. Cells were washed with 1X PBS three times for 5 minutes each. Cells were permeabilized with 0.5% Triton X-100 (T8787-100ML; Sigma-Aldrich) solution for 15 minutes, washed in PBS, and blocked with 5% normal goat serum (Cat. No. 50062Z; Thermo Fisher Scientific) for 60 minutes. Primary antibody solutions for Rabbit Anti-Vimentin (Cat. No. #5741; Cell Signaling Technology, Danvers, MA), pSTAT3$^{Ser727}$ (Cat. No. MA5-15208; Thermo Fisher Scientific), Tom70 (Cat. No. 14528-1-AP; Proteintech), and HIF1-alpha (Cat. No. NB100-134SS; Novas Biologicals) was prepared in the 1:1 PBS and normal goat serum. Cells were incubated in the primary antibody at 1:500 dilutions for 60 minutes at room temperature, then washed three times in PBS to remove the unbound antibody. Cells were then incubated with anti-Mouse/Rabbit- Alexa flour secondary antibody (Cat. No. A-11034; Thermo Fisher Scientific) solution at 1:750 dilution in PBS and normal goat serum solution for 30 minutes and then washed three times in PBS to remove the unbound secondary antibody. Next cells were stained with DAPI solution (Cat. No. 62248; Thermo Fisher Scientific) 1:2000 dilution in PBS for 30 minutes at room temperature, then washed four times with PBS for 5 minutes each. Cells were imaged with Cytation 5 fluorescence microscope (BioTek). All the images were analyzed on cells of the leading edge or in the scratch repair zone and processed with Gen5 software (BioTek).

## Multiplex for chemokines and cytokines

Multiplex cytokines and chemokines were performed using the Bio-plex kits per manufacturer instructions (Bio-RAD; Hercules, CA).

## Seahorse

Cellular oxidative phosphorylation (OXPHOS) and glycolysis were measured using the Seahorse Bioscience Extracellular Flux Analyzer (XFe96, Seahorse Bioscience Inc., North Billerica, MA, USA) by measuring oxygen consumption rate (OCR; indicative of respiration) and extracellular acidification rate (ECAR; indicative of glycolysis) in real time according to manufacturer's protocol. Spare respirator capacity (SRC) was calculated by subtracting the basal OCR from the maximal OCR.

Analysis was performed as has been previously described [17]. Briefly, 10,000 fibroblasts from healthy volunteers, keloid patients or STAT3 loss of function patients were seeded in 96-well cell culture microplates designed for XFe96 in 200 μl of appropriate growth media. Fibroblasts were cultured with various stimuli for 2 hours. Prior to measurements, growth media was removed and replaced with 180 μl pH ready Seahorse Assay Media (Agilent; Catalog #103575–100) and incubated in the absence of CO2 for 1 hour in the Biotek Cytation1 instrument during which time pre-assay brightfield images were collected. Using a Mito Stress Test assay template, cells were sequentially treated with oligomycin (2 μM), carbonyl cyanide-

4-(trifluoromethoxy)phenylhydrazone (FCCP) (0.5 μM), and rotenone + Antimycin A (0.5 μM). OCR and ECAR were then measured in a standard six-minute cycle of mix (2 min), wait (2 min), and measure (2 min). Basal levels of OCR and ECAR were recorded first, followed by OCR and ECAR levels following injection of compounds listed above. All OCR and ECAR values were normalized following the Seahorse Normalization protocol. Briefly, after the assay cells were stained with 2μg/mL Hoechst 33342 (ThermoFisher Scientific) for 30 minutes while performing post-assay brightfield imaging. Cells were then imaged and counted using the Biotek Cytation 1. Cell counts were calculated by Cell Imaging software (Agilent) and imported into Wave (Agilent) using the normalization function.

## Statistical analysis

To determine statistical significance, analysis of variance (ANOVA) with multiple-comparison corrections were applied using GraphPad Prism 8 software (San Diego, CA). Data are presented as the mean +/- SEM. A *p* value of less than 0.05 was considered significant.

## Results

### Modeled wound closure in a keloid fibroblast cell line was dependent on JAK1/2 signaling and glycolysis

Using the scratch assay and a commercially available keloid cell line, we reproduced the findings from prior reports [18–20] demonstrating an increased wound closure over time in the keloid fibroblast line compared to the healthy volunteer (HV) fibroblast line (Fig 1A). Consistent with their Warburg metabolism, the keloid fibroblast line demonstrated significant reduction in wound closure when treated with the glycolysis inhibitor 2DG but not the mitochondrial OxPhos inhibitor rotenone (Fig 1B). In contrast, the HV fibroblast line scratch closure was not significantly impacted by either treatment (Fig 1C). We next compared fibroblasts from patients with loss-of-function mutations in STAT3 to contrast against the keloid cell line with hyper-activity of STAT3. Consistent with prior reports identifying the activity and accessibility of STAT3 as central in the hyper-proliferative/hyper-motile phenotype of keloid cells [5] and similar to our prior findings [14], fibroblasts from patients with STAT3 LOF showed reduced closure in the scratch assay (Fig 1D). Keloid STAT3-mediated phenotypes in keloid cells are also known to be influenced by co-transcription with JAK2 [10]. Consistent with these reports, the accelerated wound closure in the keloid cell line was blocked by the JAK1/2 inhibitor ruxolitinib (Fig 1D) while also inhibiting closure time in the HV fibroblast line in a dose-dependent manner (Fig 1E and 1F).

### Abnormalities in metabolic balance were influenced by STAT3

Consistent with the Warburg effect, the keloid fibroblast line demonstrated: 1) more glycolytic activity as measured by extracellular acidification rate (ECAR) in the Seahorse assay (Fig 2A); 2) a significant reduction in mitochondrial ATP production (Fig 2B); 3) a non-significant reduction in basal OxPhos as measured by the oxygen consumption rate (OCR; Fig 2C and 2D); but 4) a higher spare respiratory capacity (SRC; Fig 2D and 2E). Paradoxically, despite reduced EMT in the scratch assay, fibroblasts from patients with STAT3 LOF also demonstrated increased ECAR (Fig 2A), a significant increase in SRC (Fig 2E), but without the basal abnormalities in OCR or mitochondrial ATP production (Fig 2B–2D). Treatment with ruxolitinib inhibited mitochondrial ATP production and trended towards reduced basal OCR (Fig 2C and 2D), consistent with the reported role for JAK2/STAT3 in OxPhos [21, 22]. However,

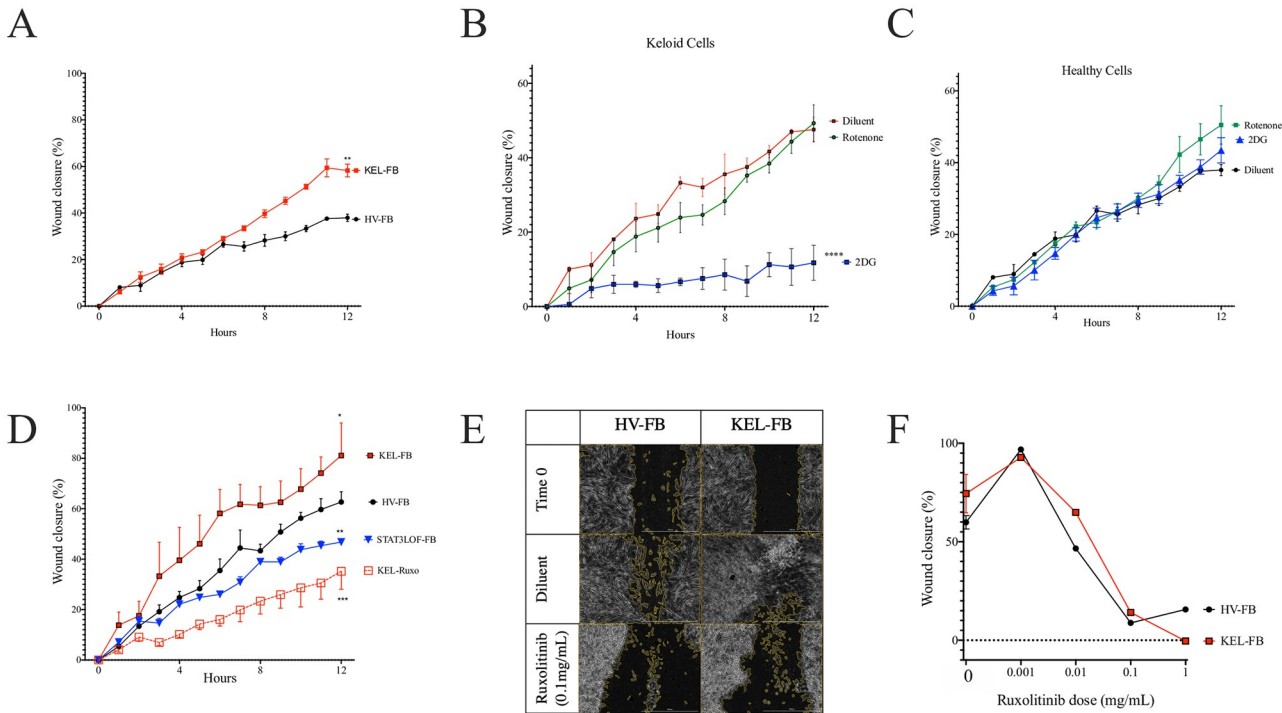

**Fig 1. EMT in keloid fibroblasts was dependent on JAK signaling and glycolysis.** (A) Wound closure over time for keloid (KEL-) and healthy volunteer (HV-) fibroblast cell lines (FB) in the scratch assay of wound repair. (B and C) Scratch assay results after treatment with the glycolysis inhibitor 2DG or the oxidative phosphorylation inhibitor rotenone for keloid (B) and HV (C) fibroblast cell lines. (D) Scratch wound repair in KEL-FB, HV-FB, contrasted against FB from patients with autosomal dominant hyper IgE syndrome due to STAT3 loss of function (STAT3 LOF) or KEL-FB treated with the JAK1/2 inhibitor ruxolitinib (Ruxo). Representative images (E) and quantitation at 20 hours (F) of wound closure in indicated cells with ruxolitinib treatment. Results are representative of three independent experiments and displayed as mean ± SEM for triplicate wells. $^{*}$ = p <0.05; $^{**}$ = p <0.01; $^{***}$ = p < 0.001, versus HV with diluent condition as determined by ANOVA with Sidak correction for area under the curve values from triplicate wells.

only in cells from patients with STAT3 LOF, ruxolitinib enhanced ECAR (Fig 2A) and shifted the ECAR-OCR ratio (Fig 2F).

## Keloid fibroblasts have alterations in transcriptional regulators of metabolism

HIF1α is a known master regulator of the hyper-glycolytic state associated with the Warburg effect [23] and is downstream of STAT3 in the EMT induction pathway [24]. Consistent with prior reports [25, 26], nuclear HIF1α expression was higher in the keloid fibroblast line but lower in STAT3 LOF cells compared to the healthy control line (Fig 3A and 3B). While rotenone treatment induced HIF1α nuclear localization in the HV fibroblast line, rotenone had no impact on the keloid line or STAT3 LOF cells (Fig 3A and 3B). This finding is consistent with reports suggesting normal HIF1α induction is dependent on functional nuclear STAT3 [25] and contributes to Warburg metabolism in keloid cells [6].

Previous work identified that keloid-derived cells have increased mitochondrial numbers, but abnormal morphology on electron microscopy [27]. Consistent with these reports and our finding of greater SRC in the keloid fibroblast line (Fig 2E), these cells had greater staining intensity for TOM70 (Fig 3C and 3D), a mitochondrial membrane protein suggested as the mitochondrial entry receptor for phosphorylated STAT3$^{Ser727}$ (pSTAT3$^{Ser727}$) [28]. STAT3 activity requires phosphorylation at one of the two primary phosphorylation sites, the serine at

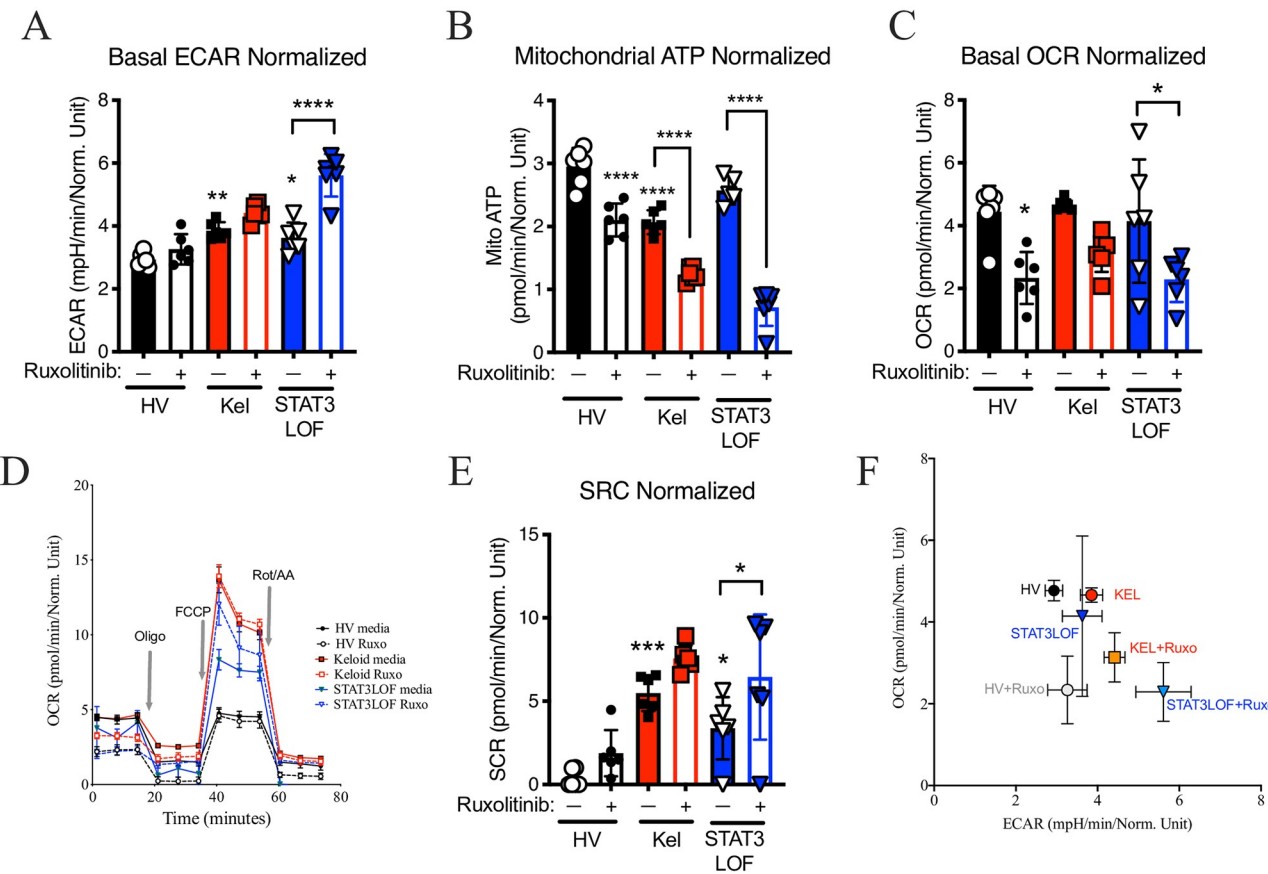

**Fig 2. Abnormalities in metabolic balance were influenced by STAT3.** Seahorse assay results for a healthy volunteer (HV) fibroblast line, a keloid (Kel) fibroblast line, or fibroblasts form patients with autosomal dominant hyper IgE syndrome due to STAT3 loss of function (STAT3LOF) with and without treatment with the JAK1/JAK2 inhibitor ruxolitinib (Ruxo): (A) basal extracellular acidification rate (ECAR), (B) mitochondrial ATP production, (C) basal oxygen consumption rate (OCR), (D) OCR tracing for indicated cell types during basal, oligomycin (Oligo) stimulation, carbonyl cyanide-4-(trifluoromethoxy)phenylhydrazone (FCCP) stimulated, and rotenone/antimycin A (Rot/AA) stimulation. (E) spare respiratory capacity (SRC), and (F) basal ECAR divided by basal OCR (ECAR/OCR ratio). Results are representative of three independent experiments and displayed as mean ± SEM for 5–6 wells per condition. * = p <0.05; ** = p <0.01; *** = p < 0.001, **** = p < 0.0001 versus HV with diluent condition unless otherwise indicated as determined by ANOVA with Sidak adjustment.

position 727 (pSTAT3$^{Ser727}$) or the tyrosine at amino acid 705 (pSTAT3$^{Tyr705}$). pSTAT3$^{Ser727}$ preferentially transits to the mitochondria through TOM70 to influence metabolism [29–31]. In contrast, pSTAT3$^{Tyr705}$ tends to transit into the nucleus and influences cell proliferation and migration [32]. Yet despite increased TOM70 and total pSTAT3$^{Ser727}$ staining, the keloid fibroblast line had a reduction in the mitochondrial associated pSTAT3$^{Ser727}$ (Fig 3C and 3E), indicating a failure of mitochondrial translocation.

### Folic acid exposure recapitulates some of the keloid phenotype in a healthy fibroblast cell line

The relative increase in pigmentation alone cannot explain the higher rate of keloid scaring in populations with darker skin [1, 33]. However, given that folate degradation is reduced in pigmented skin [34] and induces JAK2/STAT3 [35], we hypothesized that greater cutaneous folate levels may contribute to keloid pathogenesis. In healthy fibroblasts, folic acid recapitulated many of the abnormalities observed in the keloid fibroblast line including enhanced ECAR (Fig 4A); a non-significant reduction in basal OCR (Fig 4B) and mitochondrial ATP

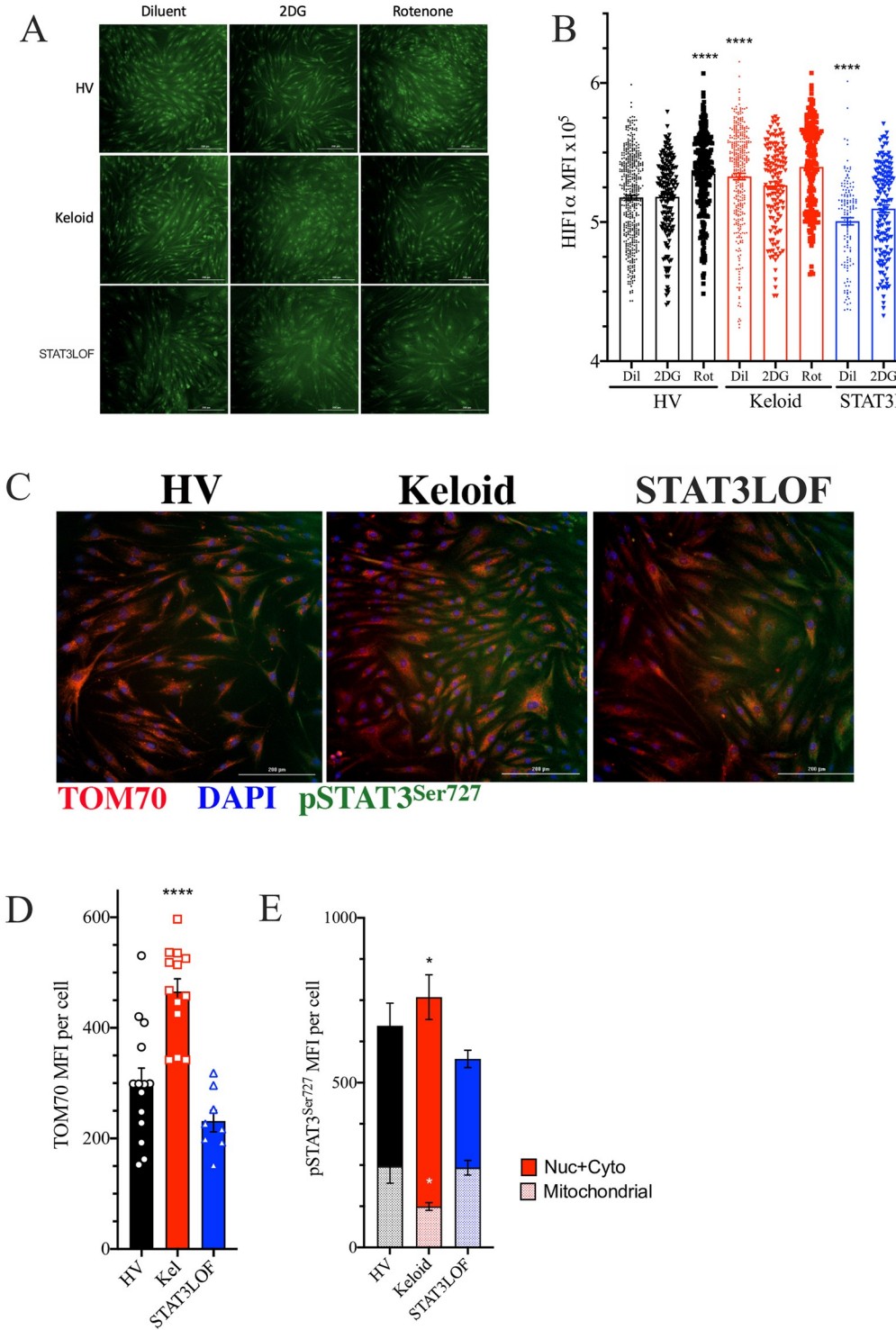

**Fig 3. Keloid fibroblasts have alterations in transcriptional regulators of metabolism.** (A and B) Immunofluorescent (IF) representative images (A) and quantitation of HIF1α staining signal co-occurring with DAPI nuclear marker in each identified cell line (B), DAPI not shown to allow nuclear localization assessment. (C) Representative IF images for cells stained for TOM70 (red), phosphorylated STAT3$^{Ser727}$ (pSTAT3$^{Ser727}$; green), and DAPI (blue). (D and E) Signaling intensity for TOM70 (D) as well as pSTAT3$^{Ser727}$ associated with TOM70 (mitochondrial) or not associated with TOM70 (nuclear and cytoplasmic; Nuc + Cyto) (E) per cell. Results are representative of two independent experiments and displayed as mean ± SEM for triplicate wells per condition. * = p <0.05; **** = p < 0.0001 versus HV with diluent condition as determined by ANOVA with Sidak adjustment.

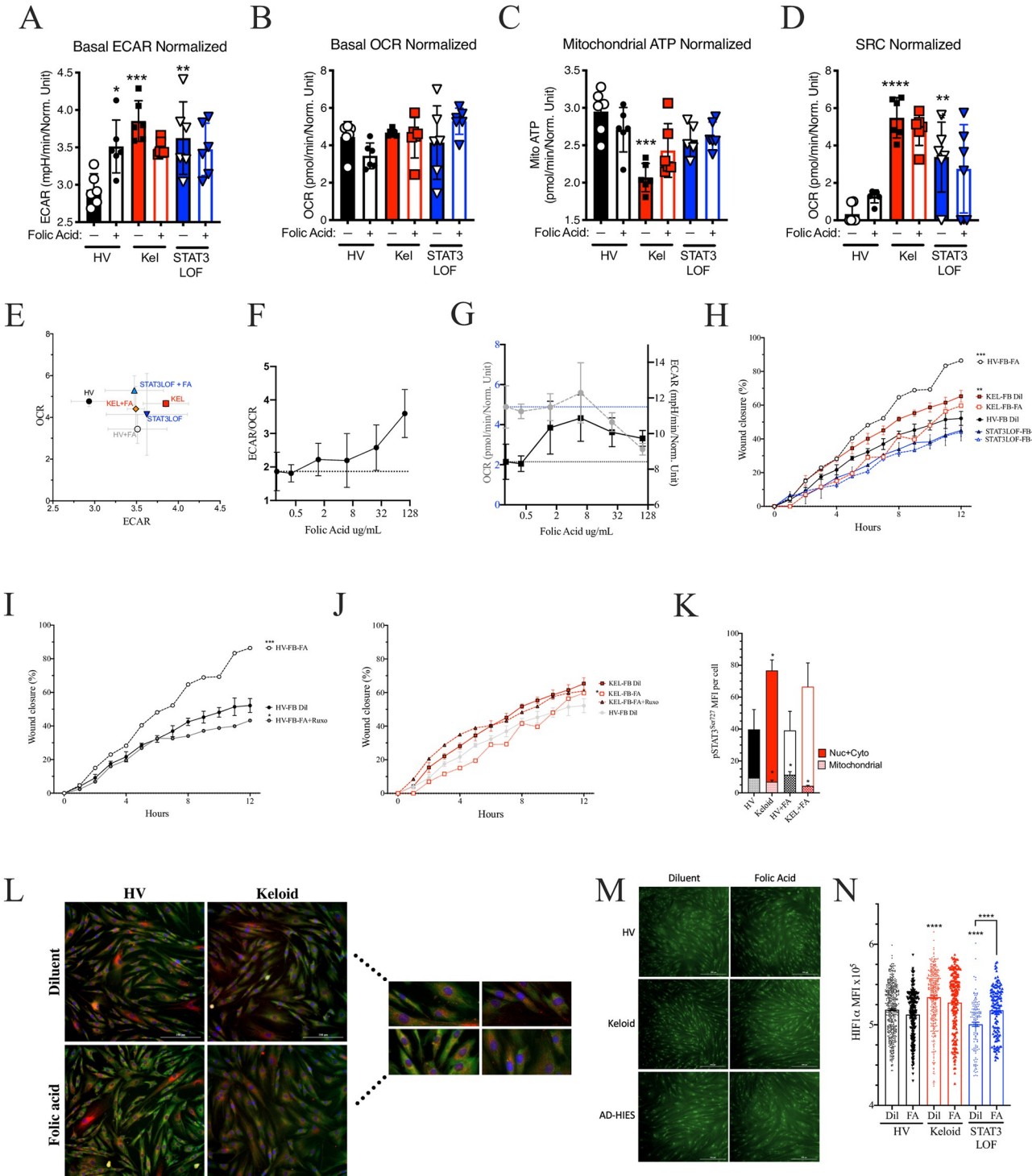

**Fig 4. Folic acid exposure recapitulates some of the keloid phenotype in healthy fibroblasts.** Seahorse assay results for a healthy volunteer (HV) fibroblast (FB) line, a keloid (Kel) fibroblast line, or fibroblasts form patients with autosomal dominant hyper IgE syndrome due to STAT3 loss of function (STAT3LOF) with and without treatment with 50ug/mL folic acid: (A) basal extracellular acidification rate (ECAR), (B) basal oxygen consumption rate (OCR), (C) mitochondrial ATP production, (D) spare respiratory capacity (SRC), and (E) basal ECAR to OCR ratio are shown. (F-G) basal ECAR to OCR ratio (F), OCR (grey, dotted lines) and ECAR (black, solid lines) raw values (G) for HV fibroblasts pre-treated with increasing doses of folic acid. (H) Scratch assay wound closure over time for indicated fibroblasts (FB) with and without 100μg/mL of folic acid (FA), (I and J) FB treated with folic acid (FA) with and without 50ug/mL of ruxolitinib (Ruxo) for the healthy (I) or keloid derived (J) FB cell lines. (K) Signal intensity for phosphorylated STAT3$^{Ser727}$ (pSTAT3$^{Ser727}$) associated with TOM70 staining (mitochondrial) or not associated with TOM70 (nuclear and cytoplasmic;

Nuc + Cyto). (L) Representative image and close-up (inset) for co-stain for TOM70 (red), pSTAT3$^{Ser727}$ (green), and DAPI (blue) as enumerated in K. (M and N) representative images (M) and quantitation of HIF1α staining signal co-occurring with DAPI nuclear marker in each identified cell (N). Results are representative of three independent experiments and displayed as mean ± SEM for 5–6 wells per condition (A–G) or triplicate wells (H–K, N). $^{*}$ = p <0.05; $^{**}$ = p <0.01; $^{***}$ = p < 0.001, $^{****}$ = p < 0.0001 versus HV with diluent condition unless otherwise indicated as determined by ANOVA with Sidak adjustment.

production (Fig 4C), but did not change in SRC (Fig 4D). Overall, folic acid shifted the ECAR-OCR ratio towards glycolysis in a dose-dependent manner (Fig 4E–4G), consistent with prior reports demonstrating glycolysis enhancement by folic acid [36, 37]. In the HV fibroblast line, folic acid enhanced EMT in the scratch assay (Fig 4H, S1A Fig) in a JAK1/ 2-dependent manner (Fig 4I). In direct opposition, in the keloid line, folic acid induced a non-significant reduction in ECAR (Fig 4A) with a non-significant enhancement of OCR and mito-chondrial ATP production (Fig 4B and 4C). In keloid fibroblasts, the ability of folic acid to inhibit wound closure in the scratch assay but was negated by JAK1/2 blockade (Fig 4H and 4J, S1A Fig). Folic acid had no significant impact on fibroblasts derived from patients with STAT3 LOF in the scratch assay (Fig 4H) or Seahorse assays (Fig 4A–4D).

While HV fibroblasts treated with folic acid recapitulated most of the keloid phenotype, folic acid treatment did not impact the mitochondrial association of pSTAT3$^{Ser727}$ (Fig 4K and 4L), nor nuclear HIF1α expression (Fig 4M and 4N) but did inhibited TGFβ production (S1B– S1D Fig). Whereas, in the keloid fibroblast line, folic acid further reduced mitochondrial pSTAT3$^{Ser727}$ (Fig 4K and 4L), without altering TGFβ production (S1B–S1D Fig) or nuclear HIF1α expression (Fig 4M and 4N). Interestingly, in STAT3 LOF fibroblasts with mutant STAT3, folic acid had far fewer impacts, including only a non-significant increase in basal OCR (Fig 4B) and a significant increase in an otherwise deficient nuclear HIF1α expression (Fig 4M and 4N). Taken together, our results suggest that folic acid mediated effects on glycolysis and mitochondrial ATP production are STAT3 dependent and disrupted in the available keloid fibroblast cell line.

## Discussion

Our findings add to the growing literature detailing the interdependence of immunity, tissue repair, and metabolism [4, 7, 14, 25, 38–40]. We demonstrate that the mutational or pharma-cologic blockade of JAK1/2 or STAT3 signaling in fibroblasts alters cellular migration/prolifer-ation, glycolysis, oxidative phosphorylation, as well as response to folic acid. In our current work, treatment with folic acid provided the greatest distinction between fibroblast cell lines from healthy volunteers and keloid scars. In the healthy fibroblast line, folic acid recapitulated many of the abnormalities in keloid fibroblasts while demonstrating opposing effects on metabolism and EMT in the cell line that was keloid derived. These findings are consistent with recent reports demonstrating efficacy for the folate targeting drug 5-FU against keloid scars [41].

We hypothesize that increased folate in pigmented skin may contribute to epigenetic changes in STAT3, consistent with prior reports [42, 43]. Alterations of mitochondrial STAT3 could then lead to a compensatory increase in STAT3, resulting in the documented increased nuclear pSTAT3$^{Tyr705}$ translocation [5], HIF1α expression [23], glycolytic activity [23], and EMT [18–20]. However, genetic alterations in STAT3 function, such as seen in STAT3 LOF, or JAK1/2 inhibition with ruxolitinib may preclude these consequences. In fact, exposure to ruxolitinib appears to further reduce mitochondrial OxPhos activity in the keloid fibroblast line and may reduce ATP production efficiency. The impacts of mitochondrial associated pSTAT3$^{Ser727}$ could not explain all of our observed differences between the keloid and HV

fibroblast lines and therefore the ultimate impact of the reduced mito-pSTAT3$^{Ser727}$ remains to be elucidated. However, our report appears to be the first to identify a baseline defect in pSTAT3$^{Ser727}$ mitochondrial localization despite the previously described increase in total pSTAT3$^{Ser727}$ [5]. Furthermore, folic acid physiology cannot present a unifying hypothesis because it only recapitulated some of the abnormalities seen in our chosen cell lines and we originally hypothesized that it would suppress mitochondrial pSTAT3$^{Ser727}$. However, our findings were successful in uncovering that the STAT3-mediated impacts of folic acid influence the mitochondrial translocation of pSTAT3$^{Ser727}$ in ways that differ between HV, keloid, and STAT3LOF fibroblast lines.

Our findings are limited by our focus on monolayer cultures of fibroblasts as the lone proxy for keloid pathology. Furthermore, our findings were derived from singular commercial cell lines. While the use of commercial cell lines may afford other researchers an opportunity to reproduce and expand upon our findings, our results cannot address the likely person-to-person, or body site-to-body site variation in healthy donors or patients with keloids. Full elucidation of keloid metabolism may also need to employ additional models and/or evaluate pathology in keloid-derived keratinocytes and melanocytes [44]. Furthermore, while our findings reproduce the findings of prior reports identifying JAK1/2, STAT3, and folate as targetable pathways for the treatment of keloids, a potentially contrasting role for vitamin D impacting STAT3 signaling and risk of keloids in pigmented skin should also be considered [34, 45].

Despite these limitations, our findings support the consideration of future clinical assessment of ruxolitinib in the treatment of keloid scars. Ruxolitinib was initially approved for the treatment of myelofibrosis, a hyper-proliferative disorder which has also been associated with abnormalities in EMT [46] and folate metabolism [47]. Moreover, our findings are consistent with other groups identifying that JAK1/2 signaling blockade reverses abnormal phenotypes in keloid-derived cells [5, 10, 11, 48]. Therefore, ruxolitinib as either an intralesional adjunct to steroids or in currently experimental topical formulations [49] may offer therapeutic benefit to patients with keloid scars if our findings can be validated in a larger number of patient-derived cell lines. However, large-scale studies of keloid scars will likely rely on surgically resected tissue given that experimental tissue collection would be unethical in this patient population.

Overall, our findings reproduce the claims of many prior reports, suggest a potential role of folate metabolism in keloid pathogenesis, and present some of the first insights into the metabolic consequences of monogenic loss of STAT3 activity. Our work expands on the current literature to directly link JAK1/2, STAT3, and HIF1α to keloid pathology, folic acid responses, and Warburg physiology; however, other signaling pathways beyond these identified pathways are likely involved for each finding. We suggest that future work assessing the global overlap between metabolic and immunologic pathways may identify additional therapeutic options for diseases marked by abnormalities in immune function, tissue repair, and metabolism.

## Supporting information

**S1 Fig. Folic acid does not impact TGFβ production.** (A) Representative scratch assay image after 12 hours for fibroblasts (FB) cell lines from a healthy volunteer (HV-) or keloid scar (KEL-) treated with folic acid (FA) and ruxolitinib (Ruxo). Masking performed by Scratch App (BioTek). (B and C) Supernatant levels for TGFβ1, 2, and 3 for FB treated with folic acid and ruxolitinib. Results are representative of two independent experiments and displayed as mean ± SEM for triplicate wells per condition. NS = not significant; $^{*}$ = p <0.05; $^{***}$ = p < 0.001, $^{****}$ = p < 0.0001 versus HV with diluent condition unless otherwise indicated as

determined by ANOVA with Sidak adjustment.
(PDF)

## Author Contributions

**Conceptualization:** Katelyn J. McCann, Manoj Yadav, Alexandra F. Freeman, Ian A. Myles.

**Data curation:** Katelyn J. McCann, Manoj Yadav, Ian A. Myles.

**Formal analysis:** Katelyn J. McCann, Manoj Yadav, Ian A. Myles.

**Funding acquisition:** Ian A. Myles.

**Investigation:** Katelyn J. McCann, Manoj Yadav, Mohammadali E. Alishahedani, Alexandra F. Freeman, Ian A. Myles.

**Methodology:** Katelyn J. McCann, Manoj Yadav, Ian A. Myles.

**Project administration:** Ian A. Myles.

**Resources:** Katelyn J. McCann, Ian A. Myles.

**Supervision:** Alexandra F. Freeman, Ian A. Myles.

**Validation:** Katelyn J. McCann, Alexandra F. Freeman, Ian A. Myles.

**Visualization:** Manoj Yadav, Ian A. Myles.

**Writing – original draft:** Ian A. Myles.

**Writing – review & editing:** Katelyn J. McCann, Manoj Yadav, Mohammadali E. Alishahedani, Ian A. Myles.

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
