## [Decision Letter · Decision Letter 0]

9 Feb 2021

PONE-D-21-01168

Differential responses to folic acid in an established keloid fibroblast cell line are mediated by JAK1/2 and STAT3

PLOS ONE

Dear Dr. Myles,

Thank you for submitting your manuscript to PLOS ONE. After careful consideration, we feel that it has merit but does not fully meet PLOS ONE’s publication criteria as it currently stands. Therefore, we invite you to submit a revised version of the manuscript that addresses the points raised during the review process. Please provide additional data to support your conclusions, as suggested by the reviewers, or modify the text accordingly.

We look forward to receiving your revised manuscript.

Kind regards,

Pankaj K Singh, Ph.D.

Academic Editor

PLOS ONE

Journal Requirements:

Reviewers' comments:

Reviewer's Responses to Questions

**Comments to the Author**

1. Is the manuscript technically sound, and do the data support the conclusions?

Reviewer #1: Partly

Reviewer #2: Yes

2. Has the statistical analysis been performed appropriately and rigorously? 

Reviewer #1: No

Reviewer #2: Yes

3. Have the authors made all data underlying the findings in their manuscript fully available?

Reviewer #1: Yes

Reviewer #2: No

4. Is the manuscript presented in an intelligible fashion and written in standard English?

Reviewer #1: Yes

Reviewer #2: Yes

5. Review Comments to the Author

Reviewer #1: Authors examine JAK2/STAT3 driven proliferation and metabolic dysregulation in keloid fibroblasts. Using JAK1/2 inhibitor ruxolitinib and 3 cell lines (healthy volunteer fibroblast cell line, keloid fibroblast cell line, and fibroblast cell line with STAT3 LOF) authors evaluated hyperproliferation and metabolic dysregulation by examining a) would closure; b) levels of OCR and ECAR; c) nuclear Hif1a; d) Tom20 levels; e) levels of mitochondrial pSTAT Ser727. Authors also examine whether addition of folic acid contributes to the altered proliferation and metabolic dysfunction in normal fibroblast cell line, keloid fibroblast cell line, and STAT3 LOF cell line. While it is an interesting study, experimental data does not necessarily directly support conclusions, and further experiments are required to validate the results.

Major points:

1) Authors state that Warburg metabolism is driven by JAK2/STAT3 metabolic adaptation. While in some scenarios Warburg effect could be driven by JAK/STAT, most of the literature does not describe requirement for JAK/STAT pathway for Warburg effect.

2) Authors use STAT3 LOF cell line as a model of JAK/STAT hyperactive keloid cell line, however throughout the manuscript results from STAT3 LOF cell line are distinct from the Keloid cell line, suggesting that STAT3 LOF has a metabolic phenotype that does not match the one found in Keloid cell line.

3) What is the impact of reduced mitochondrial levels of pSTAT3 S727 in keloid cell line? The OCR levels in HV and Keloid cell lines are not significantly different (Figure 1C), suggesting that levels of pSTAT3 S727 (Figure 3E) are not directly contributing to differences in the OXPHOS. If so, what is the significance of pSTAT3 S727?

4) Conclusion that folic acid mediates effects on glycolysis are based on data that looks at indirect effects on glycolysis and mitochondrial ATP production across three cell lines. Current data does not directly support the hypothesis that folic acid mediates effects on glycolysis. Authors should show if inhibiting glucose uptake or glycolysis (for example by 2DG) prevents effects on cell lines treated with folic acid.

Minor points:

Line 47:

Refs 5 and 10 show STAT3 plays a major role in the pathogenesis of keloids, but neither of the references mentions Warburg effect.

Line 157: 2DG instead of 2GD.

Reviewer #2: The manuscript entitled “Differential responses to folic acid in an established keloid fibroblast cell line are mediated by JAK1/2 and STAT3” by Myles et al submitted to PLOSOne described the potential role of JAK/STAT signaling and folate metabolism on the metabolic consequences during pathogenesis of keloid

Scars. This is a well-designed study with a clear message and has a great clinical significance. I recommend publication of this article after addressing some of the minor comments.

Comments:

• Authors may explain the method how they isolated the fibroblast cells from health volunteer.

• Figure 3 shown the immunofluorescence of HIF1a. This will be interesting if authors can validate this through western blotting of HIF1a protein.

6. PLOS authors have the option to publish the peer review history of their article (what does this mean?). If published, this will include your full peer review and any attached files.

Reviewer #1: No

Reviewer #2: No

---

## [Author Response · Author response to Decision Letter 0]

11 Feb 2021

Reviewer #1: Authors examine JAK2/STAT3 driven proliferation and metabolic dysregulation in keloid fibroblasts. Using JAK1/2 inhibitor ruxolitinib and 3 cell lines (healthy volunteer fibroblast cell line, keloid fibroblast cell line, and fibroblast cell line with STAT3 LOF) authors evaluated hyperproliferation and metabolic dysregulation by examining a) would closure; b) levels of OCR and ECAR; c) nuclear Hif1a; d) Tom20 levels; e) levels of mitochondrial pSTAT Ser727. Authors also examine whether addition of folic acid contributes to the altered proliferation and metabolic dysfunction in normal fibroblast cell line, keloid fibroblast cell line, and STAT3 LOF cell line. While it is an interesting study, experimental data does not necessarily directly support conclusions, and further experiments are required to validate the results.

Major points:

1) Authors state that Warburg metabolism is driven by JAK2/STAT3 metabolic adaptation. While in some scenarios Warburg effect could be driven by JAK/STAT, most of the literature does not describe requirement for JAK/STAT pathway for Warburg effect.

We have amended the text throughout that JAK/STAT is an inducer of Warburg metabolism, but is not the lone pathway of importance (most notably in the introduction, lines 47-49). We continue to highlight that one of the described master regulators of Warburg metabolism (HIF1a) is highly linked to JAK/STAT signaling. We have also clarified in the text of the discussion that other signaling pathways may play a role (lines 293-294).

2) Authors use STAT3 LOF cell line as a model of JAK/STAT hyperactive keloid cell line, however throughout the manuscript results from STAT3 LOF cell line are distinct from the Keloid cell line, suggesting that STAT3 LOF has a metabolic phenotype that does not match the one found in Keloid cell line.

We have clarified in the text that the STAT3 LOF cells are used to contrast against keloid cells – not to model them (lines 161-162). The keloid cells represent over-active STAT3 while the STAT3 LOF (loss of function) cells represent under-active STAT3. Thus, the cells lines’ distinctions are expected.

3) What is the impact of reduced mitochondrial levels of pSTAT3 S727 in keloid cell line? The OCR levels in HV and Keloid cell lines are not significantly different (Figure 1C), suggesting that levels of pSTAT3 S727 (Figure 3E) are not directly contributing to differences in the OXPHOS. If so, what is the significance of pSTAT3 S727?

We have clarified in the text that the ultimate importance of pSTAT3Ser727 remains unclear (lines 256-257). We identified a baseline difference but that difference could not explain the entire phenotype. We opted against omitting our finding because (to our knowledge) it is the first direct evidence of a mito-STAT3 defect in keloid cells even if it does not explain keloid pathogenesis. Thus we acknowledge that we have more to elucidate as to the significance of our finding but the novelty of the finding is worthy of communicating to the scientific community. 

4) Conclusion that folic acid mediates effects on glycolysis are based on data that looks at indirect effects on glycolysis and mitochondrial ATP production across three cell lines. Current data does not directly support the hypothesis that folic acid mediates effects on glycolysis. Authors should show if inhibiting glucose uptake or glycolysis (for example by 2DG) prevents effects on cell lines treated with folic acid.

The experiment the reviewer is describing would assess if the effects of folic acid were mediated by glycolysis. However, we are asserting that the effects of folic acid are mediated by STAT3 and JAK1/2 – as indicated by the loss of folic acid effects in STAT3LOF cells and cells treated with JAK1/2 inhibition. We have attempted to clarify in the text. We have also cited publications that have previously established the effects of folic acid on induction of glycolysis (lines 216-217 and new references 36 and 37). 

Minor points:

Line 47:

Refs 5 and 10 show STAT3 plays a major role in the pathogenesis of keloids, but neither of the references mentions Warburg effect.

We have amended the text to correct the descriptions. STAT3 in these papers is linked to keloid pathology, but separate work links STAT3 and Warburg physiology. We hope that our work now directly links the immunologic and metabolic pathways in STAT3.

Line 157: 2DG instead of 2GD.

This has been amended, thank you.

Reviewer #2: The manuscript entitled “Differential responses to folic acid in an established keloid fibroblast cell line are mediated by JAK1/2 and STAT3” by Myles et al submitted to PLOSOne described the potential role of JAK/STAT signaling and folate metabolism on the metabolic consequences during pathogenesis of keloid

Scars. This is a well-designed study with a clear message and has a great clinical significance. I recommend publication of this article after addressing some of the minor comments.

We thank the review for their support, insights, and time.

Comments:

• Authors may explain the method how they isolated the fibroblast cells from health volunteer.

We have clarified that the HV fibroblasts were also from cell lines purchased from ATCC. We opted to use a cell line to represent this group as it would improve reproducibility (other researchers can test these exact cells in their own lab).

• Figure 3 shown the immunofluorescence of HIF1a. This will be interesting if authors can validate this through western blotting of HIF1a protein.

We have attempted to clarify in the methods that since our findings are in the cells undergoing EMT/proliferation/migration – those cells are limited to the leading edge of the scratch. Western blotting would lack the ability to localize the expression of any protein and could only represent a lysate of the entire well. Furthermore, given that immunofluorescence is an established equal in its ability to perform protein quantitation as Western blotting, the need to assess only the cells in the scratched area indicates that the data as presented is the valid presentation.

---

## [Decision Letter · Decision Letter 1]

18 Feb 2021

Differential responses to folic acid in an established keloid fibroblast cell line are mediated by JAK1/2 and STAT3

PONE-D-21-01168R1

Dear Dr. Myles,

We’re pleased to inform you that your manuscript has been judged scientifically suitable for publication and will be formally accepted for publication once it meets all outstanding technical requirements.

Kind regards,

Pankaj K Singh, Ph.D.

Academic Editor

PLOS ONE

Additional Editor Comments (optional):

Reviewers' comments:

Reviewer's Responses to Questions

**Comments to the Author**

1. If the authors have adequately addressed your comments raised in a previous round of review and you feel that this manuscript is now acceptable for publication, you may indicate that here to bypass the “Comments to the Author” section, enter your conflict of interest statement in the “Confidential to Editor” section, and submit your "Accept" recommendation.

Reviewer #1: All comments have been addressed

Reviewer #2: All comments have been addressed

2. Is the manuscript technically sound, and do the data support the conclusions?

Reviewer #1: Yes

Reviewer #2: Partly

3. Has the statistical analysis been performed appropriately and rigorously? 

Reviewer #1: Yes

Reviewer #2: Yes

4. Have the authors made all data underlying the findings in their manuscript fully available?

Reviewer #1: Yes

Reviewer #2: Yes

5. Is the manuscript presented in an intelligible fashion and written in standard English?

Reviewer #1: Yes

Reviewer #2: Yes

6. Review Comments to the Author

Reviewer #1: (No Response)

Reviewer #2: (No Response)

7. PLOS authors have the option to publish the peer review history of their article (what does this mean?). If published, this will include your full peer review and any attached files.

Reviewer #1: No

Reviewer #2: No

---

## [Editor Report · Acceptance letter]

22 Feb 2021

PONE-D-21-01168R1 

Differential responses to folic acid in an established keloid fibroblast cell line are mediated by JAK1/2 and STAT3 

Dear Dr. Myles:

I'm pleased to inform you that your manuscript has been deemed suitable for publication in PLOS ONE. Congratulations! Your manuscript is now with our production department. 

Kind regards, 

on behalf of

Dr. Pankaj K Singh 

Academic Editor

PLOS ONE